# Interplay of Gut Microbiota, Biologic Agents, and Postoperative Anastomotic Leakage in Inflammatory Bowel Disease: A Narrative Review

**DOI:** 10.3390/ijms26157066

**Published:** 2025-07-22

**Authors:** Alexandra-Eleftheria Menni, Evdoxia Kyriazopoulou, Eleni Karakike, Georgios Tzikos, Eirini Filidou, Katerina Kotzampassi

**Affiliations:** 1Department of Surgery, Aristotle University of Thessaloniki, 54636 Thessaloniki, Greece; tzikos_giorgos@outlook.com (G.T.); kakothe@yahoo.com (K.K.); 22nd Department of Critical Care Medicine, Medical School, National and Kapodistrian University of Athens, 11527 Athens, Greece; ekyriazopoulou@windowslive.com; 3Department of Critical Care Medicine, School of Nursing, National and Kapodistrian University of Athens, 11527 Athens, Greece; elkarakike@gmail.com; 4Novo Nordisk Foundation Center for Stem Cell Medicine, reNEW, Faculty of Health and Medical Sciences, University of Copenhagen, 2200 Copenhagen N, Denmark; eirini.filidou@sund.ku.dk

**Keywords:** gut microbiota, dysbiosis, inflammatory bowel disease, *Faecalibacterium prausnitzii*, anastomotic leak, infliximab, ustekinumab, vedolizumab, adalimumab

## Abstract

Disruption of the microbiota resulting in pathogenicity is known as dysbiosis and is key in the pathogenesis of inflammatory bowel disease [IBD]. The microbiome of patients with IBD is characterized by depletion of commensal bacteria, in particular *Bacteroidetes* and the *Lachnospiraceae* subgroup of *Firmicutes,* and by the concomitant increase in *Proteobacteria* and the *Bacillus* subgroup of *Firmicutes*. These changes reflect a decrease in microbial diversity with a concomitant decrease in health-promoting bacteria like *Faecalibacterium* and *Roseburia*. Treatment with biologic agents has changed the natural course of disease, improving patient outcomes. Changes in gut microbiota occur under treatment with biologic agents and act towards reversal of dysbiosis. These changes are more striking in patients achieving remission and specific gut microbiota signatures may be predictive of treatment response and a step towards precision medicine, since, despite advances in medical treatment, some patients are at risk of surgery and subsequent complications such as anastomotic leakage. This review summarizes current available evidence on the interplay of gut microbiota and biologic agents, surgery, and surgical complications in patients with IBD.

## 1. Introduction

Inflammatory bowel disease (IBD) encompasses a variety of intestinal conditions that lead to long-lasting inflammation in the digestive system. The primary forms of IBD include ulcerative colitis (UC) and Crohn′s disease (CD), and their incidence is increasing worldwide [1,2]. IBD is thought to occur in people with a genetic predisposition following environmental exposure; gut epithelial barrier defects, the microbiota, and a dysregulated immune response are strongly implicated. Patients usually present with bloody diarrhoea, abdominal pain, weight loss, and fatigue, and the diagnosis is based on a combination of clinical and laboratory findings, such as abnormalities in the complete blood count (leucocytosis, anaemia, and thrombocytosis), elevated C-reactive protein (CRP), elevated faecal calprotectin, hypoalbuminemia, and micronutrient deficiencies. Endoscopy remains the diagnostic gold standard. The goal of medical treatment is to achieve a swift clinical response and to sustain clinical remission. Management of IBD has changed considerably in parallel with newly available therapies, and now consists of immunosuppressive agents as well as biologicals. In some patients, surgical treatment may still be required [1,2].

The gut microbiome is considered to play a key role in the pathogenesis of IBD as well as in the prediction and follow-up of treatment response. More precisely, the gut microbiome is increasingly recognized as a central factor in the development and progression of inflammatory bowel disease (IBD). It is believed to influence both the initiation and perpetuation of the disease through complex interactions with the host immune system, intestinal epithelial cells, and environmental triggers. Furthermore, alterations in the composition and function of the gut microbiota are being explored not only as potential biomarkers for diagnosing IBD but also for predicting patients’ responses to various therapeutic interventions. Consequently, monitoring the microbiome over time may offer valuable insights into treatment efficacy and disease remission or relapse, thereby playing a critical role in both personalized treatment strategies and long-term disease management. In this review, we aim to summarize current available evidence on the interplay between the gut microbiome with biologic agents, surgery, and surgical complications in patients with IBD.

## 2. Materials and Methods

A search was conducted across MEDLINE [PubMed] on 1 June 2025 under the terms “Surgical Procedures, Colorectal”[Mesh], “Anastomosis, Surgical”[Mesh], “gut microbio*”, “faecalibacterium”, “dysbiosis”, “infliximab”, “adalimumab”, “ustekinumab”, and “vedolizumab” for studies published in English without publication time restriction. The search was performed by two independent authors and the same authors assessed all articles generated by the electronic search, by title, abstract, and complete text, to find those meeting the eligibility criteria. Publications retrieved from electronic databases were imported into reference management software [EndNoteX6, Thomson Reuters, New York, NY, USA], and duplicates were removed. The search strategy is shown in Figure 1.

## 3. Results and Discussion

### 3.1. The Gut Microbiota in Inflammatory Bowel Disease

#### 3.1.1. Dysbiosis

The gut microbiota, as the endogenous gastrointestinal microbial flora is called, plays a fundamentally important role in health and disease, yet it is not completely defined and understood. There is significant intersubject variability and differences between the stool and mucosa community composition, partially attributable to variations in time, diet, and health status [3]. In IBD, major changes in the composition of the gut microbiota, compared to that of healthy individuals, have already been described. Frank and colleagues examined tissue samples from patients with Crohn′s disease (CD) and ulcerative colitis (UC), along with samples from non-inflammatory bowel disease (IBD) controls [4]. The microbiota of individuals with IBD was found to have a reduced abundance of beneficial bacteria, especially those from the *Bacteroidetes* phylum and the *Lachnospiraceae* family of *Firmicutes*, while showing a corresponding increase in *Proteobacteria* and the *Bacillus* group of *Firmicutes*. Moreover, the small-intestine samples, as a whole, contained less overall phylogenetic diversity of *Bacteroidetes* and *Firmicutes* than did the large intestine. These changes reflect a decrease in microbial diversity, both alpha-diversity (the number of different species found as a measure of local species richness) and beta-diversity (the ratio between regional and local species diversity), whereas other studies have found a concomitant decrease in health-promoting bacteria like *Faecalibacterium* and *Roseburia* [5,6]. Both bacteria of the Firmicutes phylum are producers of the short-chain fatty acid [SCFA] butyrate, and in the presence of butyrate and other SCFAs, such as propionic and acetic acid, tissue macrophages differentiate and display enhanced antimicrobial activity [7]. Most of the studies on microbiota in IBD report separately and focus on changes seen in *Faecalibacterium prausnitzii* abundance; thus, these changes are provided in more detail below.

The alteration of microbiota composition that leads to pathogenicity is referred to as dysbiosis, and it is suggested that this may be influenced by interactions with environmental stressors in individuals genetically predisposed to develop IBD [8]. Since the intestinal epithelial layer is crucial for maintaining the symbiotic relationship between microbiota and host and for preventing inappropriate immune and inflammatory reactions, this connection is disrupted in IBD. The mucosal barrier becomes permeable due to changes in epithelial tight junctions, cell death, and damage to the intestinal lining, allowing the invasion of pathogenic microorganisms in a dysbiotic setting [8].

#### 3.1.2. Faecalibacterium Prausnitzii

*F. prausnitzii* is one of the largest butyrate producers. Initially classified as Bacteroides *prausnitzii* in 1937, it was reclassified as *Fusobacterium prausnitzii* in 1974 and then as *Faecalibacterium prausnitzii* in 2002 [9]. It is imperative for sustaining health and luminal homeostasis and its abundance has been found to be altered across a wide range of diseases and disorders; in IBD, it tends to be diminished. *F. prausnitzii* consists of two main phylogroups. Phylogroup I was observed in 87% of healthy individuals, while it was present in less than 50% of patients with IBD. On the other hand, phylogroup II was found in over 75% of IBD patients but only in 52% of healthy individuals. Consequently, even though the main members of the *F. prausnitzii* population are present in both healthy and IBD individuals, richness is reduced in the latter and an altered phylotype distribution exists [10]. Reductions in the abundance of *Faecalibacterium* have been associated with a decline in the circulation of CD4^+^ CD8α^+^ Foxp3^+^ regulatory T [Treg] lymphocytes, as well as with an increase in various disease activity metrics [11]. Moreover, SCFA production by *F. prausnitzii* inhibits the enzyme HDAC1 in CD4^+^ T cells resulting in the downregulation of some proinflammatory cytokines, while also promoting Forkhead box P3 [Foxp3] which induces Th17/Treg balance by encouraging Th17 cells to differentiate [12]. Considering all of the above, it is understood that alterations in the composition and function of the gut microbiota may lead to alterations in gut microbiota-derived metabolites, such as bile acids, SCFA, and tryptophan metabolites, also implicated in the pathogenesis of IBD [13]. In health, SCFAs mediate diverse effects on mucosal immunity, including supporting B cell development, differentiation and expansion of regulatory T (Treg) cells, maintenance of mucosal integrity by means of inflammasome activation and IL-18 production, and on colonic dendritic cells and macrophages, which promote differentiation of Treg cells. All of these are impaired in IBD [13]. Bile acids that reach the colon are normally transformed into secondary bile acids by bacteria; in IBD, dysbiosis leads to loss of this transformation to secondary bile acids and an imbalance of primary and secondary bile acids in the colon [13]. Lastly, the major pathway for dietary tryptophan is the kynurenine pathway, with tryptophan also being used to generate serotonin and associated metabolites. Tryptophan can be metabolized by commensal microbiota to produce indoles with diverse effects on mucosal immunity and homeostasis. In IBD, dysbiosis leads to the loss of the microbial activation of tryptophan, causing increased metabolism down the kynurenine pathway [13].

In a cross-sectional study of 116 UC patients in remission, 29 first-degree relatives, and 31 healthy controls, *F. prausnitzii* was reduced in patients but also their relatives compared to controls. A low abundance of *F. prausnitzii* was also associated with less than 12 months of remission. *F. prausnitzii* increased steadily until reaching similar levels to those of controls if remission persisted, whereas it remained low if patients relapsed [14]. In a study involving UC patients who had not previously received biological therapy prior to starting anti-tumour necrosis factor [TNF] therapy, *F. prausnitzii* was more abundant at the start in those who responded positively compared to non-responders, and its levels increased during the induction therapy for responders [15]. Beneficial microbes like *F. prausnitzii* and *Roseburia* have been consistently associated with positive clinical outcomes [16].

### 3.2. Fluctuation of Gut Microbiota and Implications for Treatment Response Prediction

Recently, it has been acknowledged that gut microbiota in IBD fluctuates more than that of healthy individuals, periodically exhibiting the pattern of healthy individuals and deviating away from it. Some fluctuations of the gut microbiota correlate with disease severity, while others have been associated with intensified medication due to a flare-up of the disease, implying a future direction for microbiota composition guided therapies [17,18,19]. In a recent systematic review focusing on biomarkers for response to advanced therapies in IBD, baseline microbial analyses and therapeutic response data were available for a total of 1232 individuals, making up 46% of the overall study population [20]. Parameters evaluated as biomarkers for treatment response in the individual studies were diversity, abundance of specific microbial taxa, presence of opportunistic organisms, SCFA-producing organisms and butyrate synthesis pathways, and other metabolomic analysis [20]. Evidence from individual studies suggests that dysbiosis, characterized by a decrease in Firmicutes, and/or enterotyping correlate with the risk and time to relapse after anti-TNF treatment [21,22,23]. Moreover, a range of metabolic biomarkers involving lipid, bile acid, and amino acid pathways may contribute to prediction of response to anti-TNF therapy in IBD [24]. In a proof-of-concept investigation, Busquets et al. examined the ability to predict who would respond to anti-TNF therapy. The RAID algorithm, which combines four bacterial markers, demonstrated a strong ability to differentiate between responders and non-responders, achieving sensitivity and specificity rates of 93.33% and 100%, respectively, with a positive predictive value of 100% and a negative predictive value of 75% [25].

### 3.3. Longitudinal Changes in Gut Microbiota in IBD Patients Treated with Biologic Agents

In order to achieve the double goal in IBD treatment of rapid clinical response and sustained clinical remission, a variety of immunomodulating therapies are available (corticosteroids, thiopurines and newer biologic agents such as TNF inhibitors). Anti-TNF-α agents are among the most effective therapies to induce and maintain remission in IBD, as a T-cell mediated response is the hallmark of the pathogenesis of intestinal inflammation, which is facilitated by increased proinflammatory cytokines, interferon gamma, interleukin 12, and TNF-α. Increased secretion of TNF-α from lamina propria mononuclear cells has been found in the intestinal mucosa and TNF-α positive cells have been found deeper in the lamina propria and in the submucosa [1].

As dysbiosis is also considered to play a key role in IBD, several studies so far have investigated possible changes in the composition of gut microbiota during and/or after treatment with biologic agents compared to baseline; these studies are summarized in Table 1 [15,25,26,27,28,29,30,31,32,33,34,35,36,37,38,39,40,41]. The evidence relates mainly to microbiota changes under treatment with TNF inhibitors but also with other treatments, discussed below in more detail.

#### 3.3.1. Adalimumab

Adalimumab is a TNF-blocker used in the treatment of IBD. In a prospective study in Italy, microbiota of 20 patients with CD under adalimumab was investigated prior to and after 6 months of treatment [26]. In the whole population, there was a trend towards increased abundance of some main phyla such as *Firmicutes* and *Bacteroides*, and a decreased abundance of Proteobacteria. These differences reached statistical significance only in patients with treatment success, suggesting that response to adalimumab may be associated with reversal of dysbiosis. No notable alterations were detected in the population of *Faecalibacterium prausnitzii*. Comparable findings have also been documented by other researchers in both CD [25,27] and UC [28,29].

#### 3.3.2. Infliximab

Infliximab is another anti-TNF for the treatment of IBD. In a German study, among patients with IBD and rheumatic diseases and healthy controls, Aden et al. reported significant baseline differences among the three groups; dysbiosis was present in both the IBD and rheumatic groups [30]. Interestingly, treatment with infliximab shifted the diversity of faecal microbiota in patients with IBD, but not with rheumatic disease, toward that of controls, suggesting that other factors may interfere with biologicals and induce clinical remission. Ditto et al. focused on microbiota changes under treatment with TNF-inhibitors [TNFi] in a small cohort of patients with enteropathic spondylarthritis. Although no differences were observed in α- or β-diversity, abundance of *Lachnospiraceae* and *Coprococcus* increased and there was a decreasing trend in *Proteobacteria* and *Gammaproteobacteria* [31]. Similar studies confirm the restoration of gut dysbiosis in IBD patients under infliximab treatment, reflected in the increase in α-diversity, increased abundance of *Bacteroidetes* and *Firmicutes*, and decreased abundance of *Enterobacterales,* as well as an increased abundance of SCFA-producing taxa such as *Lachnospira*, *Roseburia,* and *Blautia* [family *Lachnospiraceae*] [32]. These changes are more pronounced in treatment-responders and specific alterations like the ratio of *F. prausnitzii/E. coli* are accurate biomarkers for predicting clinical remission, performing even better than calprotectin and the Harvey–Bradshaw index [15,33,34].

#### 3.3.3. Ustekinumab

Ustekinumab is a fully human IgG1κ monoclonal antibody that targets the shared p40 subunit of interleukin [IL]-12 and IL-23. By preventing these cytokines from attaching to their receptors, it diminishes the maturation and proliferation of Th-1 and Th-17 cells. In a limited study involving patients with IBD, no significant long-term effects of ustekinumab treatment were observed on alpha- and beta-diversity, nor on the abundance of other phyla or genera [36]. In a secondary analysis of a randomized, double-blind, placebo-controlled phase 2b clinical trial, among 306 anti-TNF-refractory CD patients, a high relative abundance of *Faecalibacterium* was associated with remission 6 weeks later. The median α-diversity of responders observed after the introduction of ustekinumab showed a significant evolution over time, increasing from baseline to 4 weeks post-induction. It then declined from 4 to 6 weeks after treatment initiation, but was notably higher than the baseline at 22 weeks post-induction, indicating that changes in the microbiota may reflect treatment response [37]. Xu et al. have even reported changes in the oral microbiota under ustekinumab treatment in responders and non-responders; oral samples are easier than faecal samples to obtain for follow-up of treatment response [38].

#### 3.3.4. Vedolizumab

Vedolizumab is a human-derived IgG1 monoclonal antibody that targets α4β7 integrin, specifically inhibiting the movement of leukocytes into the gastrointestinal tract by binding to the α4β7 integrin. In a small study of 29 participants, patients in remission under anti-integrin therapy had a higher abundance of the phylum *Verrucomicrobiota* and metabolomic analysis showed higher levels of two SCFA, namely butyric acid and isobutyric acid, in these patients compared to non-responders [39]. Ananthakrishnan et al. studied, longitudinally, the gut microbiota of 85 patients with IBD under vedolizumab and observed very few changes in microbial composition under treatment [40].

Nevertheless, there were markedly greater changes in microbial function at the metagenomic level. In patients with Crohn′s disease, 17 pathways showed a significant reduction at the 14-week follow-up compared to the baseline, with 15 of these reductions occurring exclusively in individuals who reached remission. Among these were declines in various tricarboxylic acid cyclic pathways and the nicotinamide adenine dinucleotide salvage pathway, indicating a reduction in oxidative stress for those achieving remission. The changes were less striking in UC.

### 3.4. Surgery in Inflammatory Bowel Disease, Surgical Complications, and Gut Microbiota

#### 3.4.1. Surgery and the Role of Gut Microbiota

Over the last two decades, although advances have improved the course and outcomes of IBD, like earlier diagnosis, introduction of disease-modifying immunosuppressive therapy and biologic agents, earlier detection, and endoscopic management of colorectal neoplasia, there is still a considerable 5-year cumulative risk of surgery for such patients, reaching 7% in UC and 18.0% in CD [42]. Recently, Lewis et al. [43] described an association of dysbiosis and surgery for CD. They identified patients with prior surgery in two different cohorts, namely the Study of a Prospective Adult Research Cohort with Inflammatory Bowel Disease [SPARC IBD] and the Diet to Induce Remission in Crohn′s Disease [DINE-CD] study, and reported that intestinal resection was associated with reduced alpha-diversity and altered beta-diversity with increased *Proteobacteria* and reduced *Bacteroidetes* and *Firmicutes*.

Additionally, the potentially beneficial *Egerthella lenta*, *Adlercreutzia equalofaciens,* and *Gordonibacter pamelaeae* were lower in abundance among patients with prior surgery in both cohorts [43]. In a separate study examining alterations in microbiota and metabolome following various surgeries for IBD, it appears that intestinal surgery leads to a decrease in the diversity of the gut microbiota and metabolome. Colectomy had a greater effect compared to ileocolonic resection and the type of surgery explained the greater variation in the microbiota data than any other variable, followed by disease subtype, antibiotic use, and disease activity [44].

#### 3.4.2. Anastomotic Leakage and the Role of Gut Microbiota

Complications following intestinal surgery affect roughly one in three patients. The most prevalent short-term complications are infection and ileus, while pouchitis and faecal incontinence are the most commonly observed long-term complications [45]. Anastomotic leakage is a major complication, leading to high morbidity and mortality. Anastomotic leakage is quite common after colorectal surgical procedures with an incidence of 2–20%, depending on the location of the anastomosis, with the highest rates being observed in low rectal anastomoses. Common predisposing factors are mainly host-derived [male gender, increasing age, comorbidities, and malnutrition] but operation characteristics [duration of surgery, small distance of the anastomosis from the anal verge, and positive intraoperative leak test] are of paramount importance [46]. Mechanical bowel preparation and antimicrobial prophylaxis may decrease the risk [47]. The underlying pathophysiology is yet unclear but microbiota changes have been proposed as a potential contributor.

An increasing volume of research indicates that undergoing surgery can lead to a notable shift in the composition of the gut microbiota due to ischemia-reperfusion injury occurring from the constriction of intestinal blood vessels during the procedure [48]. Following colorectal surgery, the relative abundance of oral anaerobic bacteria, including *Parvimonas micra*, *Peptoanaerobacter stomatis*, *Peptostreptococcus anaerobius*, *Dorea longicatena*, and *Porphyromonas uenonis*, as well as obligate anaerobes like bifidobacteria, is significantly decreased. Conversely, the levels of pathogenic bacteria such as *Enterobacteriaceae*, *Enterococcus*, *Staphylococcus*, and *Pseudomonas* substantially increase after the procedure. [49]. Anastomotic leakage is considered to be associated with gut microbiota changes, and has already been shown in various animal models, suggesting common pathogens like *Pseudomonas aeruginosa* and *Enterococcus faecalis* as factors contributing to leakage [50,51,52]. A recent study using a mouse model suggests that gut microbiota influence anastomotic healing in colorectal surgery through modulation of mucosal proinflammatory cytokines, such as mucosal MIP-1α, MIP-2, MCP-1, and IL-17A/F [53], so that biopsy samples from surgical margins, rather than faecal samples, may be appropriate to explore the contribution of the intestinal microbiota to leakage [54].

In a small cohort of 21 colon cancer patients, five developed anastomotic leak and showed an array of bacterial species which promoted dysbiosis, such as *Acinetobacter lwoffii* and *Hafnia alvei*. Patients with appropriate mucosal healing had a microbiota abundant in species with a protective function like *Faecalibacterium prausnitzii* and *Barnesiella intestinihominis* [55]. Lehr et al. [56] concluded that after colorectal surgery, overall bacterial diversity and the abundance of some genera such as *Faecalibacterium* or *Alistipes* decreased over time, while the genera *Enterococcus* and *Escherichia_Shigella* increased.

Marked variations have been observed in the abundance of genera including *Prevotella*, *Faecalibacterium*, and *Phocaeicola. Ruminococcus2* and *Blautia* demonstrated notable differences in their abundance, even among preoperative samples, suggesting that they may also serve as predictors for postoperative complications [56]. A recent systematic review on the role of the gut microbiota in anastomotic leakage after colorectal resection summarized results from seven clinical and five experimental studies. The authors conclude that patients experiencing anastomotic leakage show a reduced α-diversity in their gut microbiota, and certain genera of microbes, such as *Lachnospiraceae, Bacteroidaceae, Bifidobacterium, Acinetobacter, Fusobacterium, Dielma, Elusimicronium, Prevotella,* and *Faecalibacterium*, appear to be linked to this condition. In contrast, genera like *Streptococcus, Eubacterium, Enterobacteriaceae, Klebsiella, Actinobacteria, Gordonibacter, Phocaeicola,* and *Ruminococcus* seem to provide a protective effect [57].

#### 3.4.3. Anastomotic Leakage and the Role of Biologic Agents

Since TNF is involved in angiogenesis and collagen production, which are crucial for wound healing, there are concerns that pre-operative treatment with anti-TNF could affect the body′s response to surgical stress and elevate the likelihood of complications during surgery. In a prospective, multi-centre cohort pilot study, 46 IBD patients undergoing major abdominal operations were analysed, with 18 of them having undergone anti-TNF therapy prior to the procedure. Concentrations of immunological and other biomarkers of the surgical stress response [TNF, IL-6, IL-10, IL-8, IL-17A, C-reactive protein, white blood cells, cortisol, transferrin, ferritin, and D-Dimer] were measured and no difference in the concentrations was found between anti-TNF-treated and anti-TNF-naïve patients postoperatively.

Moreover, no difference in the rate of postoperative complications or length of stay was observed [58]. In a retrospective study of 282 IBD patients, of whom 73 patients were treated with anti-TNF therapy within 8 weeks of surgery, 30-day anastomotic leak, intra-abdominal abscess, wound infection, extra-abdominal infection, readmission, and mortality rates did not differ significantly [59]. Similar results have been reported by others [60,61]. In a cohort of 417 surgically treated patients with CD in Demark in 2000–2007, however, prednisolone, rather than biologic agents, had a negative impact on post-surgical anastomotic leak rates [60]. In a meta-analysis of six studies including 1159 patients, among whom 413 complications were identified, there was no significant difference in the major complication rate [OR, 1.59; 95% CI, 0.89–2.86], minor complication rate [OR,1.80; 95% CI, 0.87–3.71], reoperation rate [OR, 1.33; 95% CI, 0.55–3.20], or 30-day mortality rate [OR, 3.74; 95% CI, 0.56–25.16] between the infliximab-treated and control groups [62]. Importantly, timing of the last dose of anti-TNF agents does not appear to affect the rate of postoperative complications in patients with IBD [63].

## 4. Conclusions

Dysbiosis plays a central role in the pathogenesis of IBD. Biologic therapies have significantly altered the disease course, leading to improved patient outcomes. These treatments also induce changes in the gut microbiota, and specific microbial signatures may help predict treatment response, paving the way for precision medicine. Despite therapeutic advances, some IBD patients still require surgery and remain at risk of complications such as anastomotic leakage, which has been linked to alterations in the gut microbiota [64]. To date, no studies have specifically examined the relationship between gut microbiota and anastomotic leakage in patients with IBD.

Given the evidence, biologic treatment may offer additional benefits to surgical candidates by modulating the gut microbiota toward eubiosis, potentially reducing the risk of anastomotic leakage [Figure 2]. This hypothesis warrants further investigation, and the complex interplay between gut microbiota, biologic therapy, and postoperative outcomes in IBD should be explored in future animal and clinical studies.

## Figures and Tables

**Figure 1 ijms-26-07066-f001:**
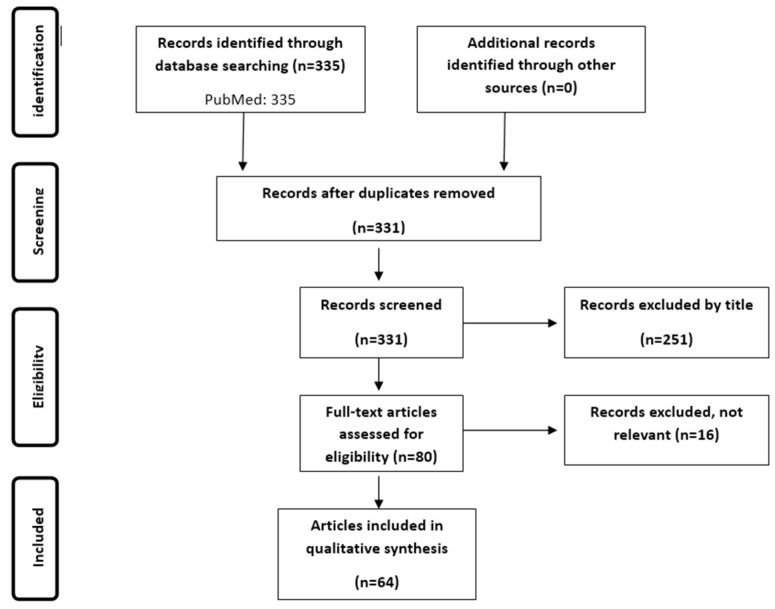
Search strategy.

**Figure 2 ijms-26-07066-f002:**
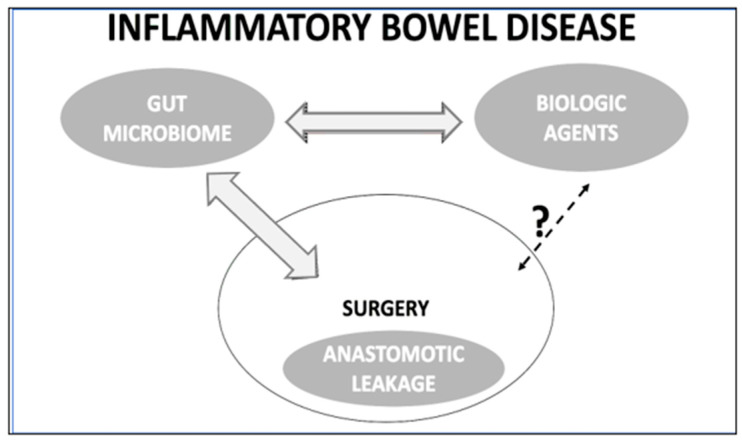
The concept of interplay among the gut microbiota, biologic agents, and the postoperative anastomotic leakage in patients with inflammatory bowel disease.

**Table 1 ijms-26-07066-t001:** Longitudinal changes in gut microbiota in patients with inflammatory bowel diseases treated with biologic agents.

Ref.	Disease	Biologic Agent	Patients	Type of Sample	Main Outcomes	*F. prausnitzii*
[15]	UC	TNF inhibitors	56	Faecal, sigmoid biopsies	Not reported	↑ especially in responders
[25]	CD	Adalimumab	15	Rectal mucosal biopsy	↑ *Firmicutes*, ↑ *Bacteroides,* and ↑ *Actinobacteria/*↓ *E. coli*	Non-significant changes
[26]	CD	Adalimumab	20	Faecal	↓ *Proteobacteria* in responders (from 15.8% to 6.8%; p: 0.049)	Non-significant changes
[27]	CD	Adalimumab	8	Faecal	No differences in α- and β-diversity ↑ *Firmicutes* and ↓ *Proteobacteria* in responders	Not reported
[28]	UC	Adalimumab	131	Faecal	↓ *Burkholderia-Caballeronia-Paraburkholderia,* ↓ *Staphylococcus*, ↑ *Bifidobacterium*, and ↑ *Dorea* in responders	Not reported
[29]	UC	Adalimumab	9	Biopsy	↑ *Dorea* and ↑ *Lachnospira* in responders	Not reported
[30]	IBD	Infliximab, vedolizumab	35	Faecal	↑ α-diversity	Not reported
[31]	IBD + EA	TNF inhibitors	20	Faecal	No differences in α- and β-diversity ↑ *Lachnospiraceae* and ↑ *Coprococcus*	Not reported
[32]	CD	Infliximab	49	Faecal	↑ α-diversity ↑ *Bacteroidetes*, ↑ *Firmicutes*, ↓ *Enterobacterales* ↑ SCFA-producing taxa (*Lachnospira*, *Blautia*)	Not reported
[33]	IBD	TNF inhibitors	27	Faecal	↑ α-diversity only in responders ↑ *Firmicutes*, ↑ *Lachnospiraceae* in responders	↑ *F. prausnitzii/E. coli* (F/E) ratio in responders
[34]	IBD	Infliximab	40	Faecal	↑ Shannon in patients with mucosal healing ↑ *Blautia,* ↑ *Bacteroides*, and ↓ *Prevotella* in patients with mucosal healing	↑ in patients with mucosal healing
[35]	UC	Infliximab, etrolizumab	287	Faecal	Shannon diversity and species richness ↑ in remitters *Bifidobacterium breve* ↓	More abundant in non-remitters
[36]	IBD	Ustekinumab	11	Faecal	No differences in α- and β-diversity No differences in abundances of phyla or genera	Not reported
[37]	CD	Ustekinumab	306	Faecal	α-diversity of responders changed over time (↑ from baseline to 4 weeks, ↓ from 4 to 6, and ↑ than baseline at 22 weeks)	More abundant at baseline in responders
[39]	UC	Vedolizumab	29	Faecal	↑ *Verrucomicrobiota* in responders	Not reported
[40]	IBD	Vedolizumab	85	Faecal	In CD, *Bifidobacterium longum, Eggerthella, Ruminococcus gnavus, Roseburia inulinivorans,* and *Veillonella parvula* ↓ in responders. In UC, *Strepotococcus salivarium* ↑ in non-responders.	Not reported
[41]	UC	Vedolizumab	45	Faecal	↑ *Firmicutes*	Not reported

Abbreviations: CD, Crohn’s disease; IBD, inflammatory bowel disease; UC, ulcerative colitis; and SCFA, short-chain fatty acids.

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
