# Peer review of "Interplay of Gut Microbiota, Biologic Agents, and Postoperative Anastomotic Leakage in Inflammatory Bowel Disease: A Narrative Review"

_ijms, 2025, doi:10.3390/ijms26157066_

Round 1
Reviewer 1 Report
Comments and Suggestions for Authors
The manuscript primarily focuses on the interplay between gut microbiota, biologic agents, and postoperative anastomotic leakage in inflammatory bowel disease. The overall writing and content are of good quality, but I have the following comments for the authors to address:
- Lines 49–52: The introduction to the role of gut microbiota in IBD is overly simplistic.
- Line 119: What are the specific types of short-chain fatty acids and tryptophan metabolites undergoing changes here?
- Line 127: The authors should confirm whether "TNF" should be "TNF-α," as TNF-α is widely recognized as a key inflammatory factor in IBD. The authors should also review the entire manuscript for consistency—for example, adalimumab is a monoclonal antibody biologic targeting TNF-α.
- Line 175: Faecalibacterium prausnitzii should be italicized.
- Line 283: The authors mention some data related to colorectal cancer patients, which seems inconsistent with the main focus of this paper.
- The authors could supplement Figure 2 with markers of altered gut microbiota and representative biologic agents.
- The conclusion section is overly complex and should be streamlined for clarity.
- The overall writing requires further review and revision by the authors.
Author Response
File attached.

Reviewer 2 Report
Comments and Suggestions for Authors
The manuscript by Alexandra-Eleftheria Menni and coworkers is a review article that aims to summarize the “Interplay of gut microbiota, biologic agents and postoperative anastomotic leakage in inflammatory bowel disease” according to its title. The Authors discuss changes in the intestinal microbiota that occur in the inflammatory bowel disease (IBD) – ulcerative colitis and Crohn’s disease. Next, they present new therapies based on monoclonal antibodies against TNFα that are described in the manuscript as “biologic agents”. Finally, they present surgical complications in relation to microbiota composition. Although some aspects of the review seem novel, i.e. the subsection presenting the link between anastomotic leakage and the role of gut microbiota, the manuscript is not easy to follow and do not provide sufficient information for readers, especially those that may not be too familiar with the subject. On the other hand, readers who are professionally involved in this area of research will find this review too general.
Detailed remarks:
- The authors use the expression “biologic agents” many times throughout the text, but in fact they focus only on therapies using different monoclonal anti-TNF antibodies. They never explain why the approach of blocking TNFα is used in treatments of IBD and what is the exact role of TNFα in the course of ulcerative colitis and Crohn’s disease. In fact, even in chapter 3.3. describing the connection between therapies and gut microbiota, detailed information about the adalimumab and infliximam are not provided, but are restricted to a general expression “a TNF-blocker” or “another anti-TNF”.
- The authors use many general descriptions throughout the text. For example in lines 42-44, when describing the principles of diagnosis, the Authors simply state that “the diagnosis is based on a combination of clinical and laboratory findings”. What exactly does it mean? What doagnostic markers are used?
- A short explanation of alpha and beta diversity should be provided the first time this phrase appears in the manuscript.
- Why the Authors focus only on butyric acid when they describe the effect of short chain fatty acids (SCFA)? All three: butyric acid, propionic acid, and acetic acid are important.
- Why the chapter 3.1.2 focuses only on Faecalibacterium prausnitzii? Readers who are unfamiliar with the subject may assume that this is the main bacteria found in the intestines, or at least the only one that shows changes and dysbiosis. There are several available review articles that describe the problem of dysbiosis in IBD much better. The present review article does not provide readers with sufficient amount of information about the changes in the composition of gut microbiota in IBD patients.
- In line 112 the Authors write about Treg cells, describing these cells as regulatory leukocytes. If they meant regulatory T lymphocytes (Tregs) these cells are most commonly detected by their expression of CD4, CD25 and FoxP3, not by CCR6 + CSCR6 + DP8α.
- In lines 117-120 the Authors state: “Considering all the above, it is understood that alterations in the composition and function of the gut microbiota may lead to alterations in gut microbiota-derived metabolites, such as bile acids, short- chain fatty acids and tryptophan metabolites, also implicated in the pathogenesis of IBD”. However, they never expand information about the role of bile acids or tryptophan metabolites.
- The sentence in lines: 261-263 seems to be missing some kind of ending or conclusion: “Common predisposing factors are mainly host-derived [male gender, increasing age, comorbidities, malnutrition] but operation characteristics [duration of surgery, small distance of the anastomosis from the anal verge, positive intraoperative leak test] [46].” But operation characteristics what ….???

There are some editorial and language mistakes:
in line 79 a full stop (.) is missing between the words “individuals” and “Frank”;
in line 268 “the makeup of the gut microbiota” should be replaced with “the composition of the gut microbiota”;
in line 269 the phrase “In summary” is unnecessary;
in line 305 the word “ worries” should be preplaced with “concerns”.
Round 2
Reviewer 1 Report
Comments and Suggestions for Authors
I don't have any other suggestions.
Reviewer 2 Report
Comments and Suggestions for Authors
The Authors have took under consideration all my remarks, followed my suggestions and corrected the manuscript accordingly. After these improvements the manuscript is easier to understand even for readers unfamiliar with the topic.
My only remark refers to the section "Patents" which has no content, thus should be deleted from the text. According to the IJMS template: "This section is not mandatory but may be added if there are patents resulting from the work reported in this manuscript."
In my opinion, the Authors have made a sufficient effort to improve their manuscript that can be recommended for publication in IJMS in the present form.